# Modeling the onset of symptoms of COVID-19: Effects of SARS-CoV-2 variant

**Joseph R. Larsen**[1,2‡], **Margaret R. Martin**[3‡], **John D. Martin**[4]*, **James B. Hicks**[2]*, **Peter Kuhn**[2]*

1 Quantitative and Computational Biology, Department of Biological Science, University of Southern California, Los Angeles, California, United States of America, 2 Convergent Science Institute in Cancer, Michelson Center for Convergent Bioscience, University of Southern California, Los Angeles, California, United States of America, 3 Department of Computer Science, Tufts University, Medford, Massachusetts, United States of America, 4 Materia Therapeutics, Las Vegas, Nevada, United States of America

‡ These authors are co-first authors on this work.
* jdmartin@alum.mit.edu (JDM); jameshic@usc.edu (JBH); pkuhn@usc.edu (PK)

**Data Availability Statement:** All relevant data are within the manuscript and its Supporting Information files. Publicly available datasets were used for this study. These can be found at: https://www.who.int/publications/i/item/report-of-the-

## Abstract

Identifying order of symptom onset of infectious diseases might aid in differentiating symptomatic infections earlier in a population thereby enabling non-pharmaceutical interventions and reducing disease spread. Previously, we developed a mathematical model predicting the order of symptoms based on data from the initial outbreak of SARS-CoV-2 in China using symptom occurrence at diagnosis and found that the order of COVID-19 symptoms differed from that of other infectious diseases including influenza. Whether this order of COVID-19 symptoms holds in the USA under changing conditions is unclear. Here, we use modeling to predict the order of symptoms using data from both the initial outbreaks in China and in the USA. Whereas patients in China were more likely to have fever before cough and then nausea/vomiting before diarrhea, patients in the USA were more likely to have cough before fever and then diarrhea before nausea/vomiting. Given that the D614G SARS-CoV-2 variant that rapidly spread from Europe to predominate in the USA during the first wave of the outbreak was not present in the initial China outbreak, we hypothesized that this mutation might affect symptom order. Supporting this notion, we found that as SARS-CoV-2 in Japan shifted from the original Wuhan reference strain to the D614G variant, symptom order shifted to the USA pattern. Google Trends analyses supported these findings, while weather, age, and comorbidities did not affect our model's predictions of symptom order. These findings indicate that symptom order can change with mutation in viral disease and raise the possibility that D614G variant is more transmissible because infected people are more likely to cough in public before being incapacitated with fever.

## Author summary

We developed a mathematical model to predict symptom order of symptomatic COVID-19 cases from patient characteristics data in the USA and China. Surprisingly, our model predicted that cough occurs first in the USA, while fever occurs first in China. We

who-china-joint-mission-on-coronavirus-disease-2019-(covid-19) https://www.cdc.gov/mmwr/volumes/69/wr/mm6924e2.htm https://www.gastrojournal.org/article/S0016-5085(20)30448-0/fulltext https://www.nature.com/articles/s41562-020-0928-4 https://www.mdpi.com/2077-0383/9/9/2925 https://academic.oup.com/cid/advance-article/doi/10.1093/cid/ciaa1470/5912544 https://www.journalofinfection.com/article/S0163-4453(20)30119-5/fulltext https://www.nejm.org/doi/full/10.1056/NEJMc2010419 https://www.cdc.gov/mmwr/volumes/69/wr/mm6925e1.htm https://erj.ersjournals.com/content/55/5/2000547 https://jamanetwork.com/journals/jamanetworkopen/fullarticle/2767216 https://www.thelancet.com/journals/lanonc/article/PIIS1470-2045%2820%2930310-7/fulltext https://www.nature.com/articles/s41591-020-0979-0 https://www.thelancet.com/journals/eclinm/article/PIIS2589-5370(20)30259-5/fulltext https://academic.oup.com/jid/advance-article/doi/10.1093/infdis/jiaa380/5864898 Code for this study can be found at: https://github.com/j-larsen/Stochastic_Progression_of_COVID-19_Symptoms

**Funding:** We acknowledge funding support by the Dr. Peter N. Schlegel, M.D., Family Endowed Fellowship Fund awarded to JRL; Hsieh Family Foundation and Kathy & Richard Leventhal Research Fund awarded to PK. The funders had no role in study design, data collection and analysis, decision to publish, or preparation of the manuscript.

**Competing interests:** I have read the journal's policy and the authors of this manuscript have the following competing interests: JDM is employed by Materia Therapeutics. The remaining authors have declared that no competing interests exist.

hypothesized the difference is due to the SARS-CoV-2 D614G variant, which was predominate in the USA during data collection, whereas the original Wuhan reference strain was predominate in China. To test this, we used patient data from the outbreak in Japan, which was initially dominated by the Wuhan reference strain but eventually dominated by the D614G variant. The predicted symptom order changed with the viral variant, but not region, weather, patient age, or comorbidity. These results support the notion that cough occurs earlier in the D614G variant than the Wuhan reference strain. The D614G variant's greater transmissibility might be explained by infected individuals coughing and spreading the virus before they are incapacitated by fever. Additionally, we hope other researchers will further investigate symptom order of infectious diseases to understand how viral variants and comorbidities affect disease progression. Such work is especially important now as contagious and deadly variants of SARS-CoV-2 are under investigation and rapidly spreading worldwide.

## Introduction

The Coronavirus Disease 2019 (COVID-19) is a global pandemic that has reached over two hundred million confirmed cases worldwide as of October 12, 2021 [1]. As cases continue to rise globally [2], understanding of the severe acute respiratory syndrome coronavirus 2 (SARS-CoV-2) and how it transmits has increased. Direct person-to-person contact has been determined to be the most common mode of transmission through respiratory droplets spread from asymptomatic people and the coughs of symptomatic people [3]. Societies can use various methods to reduce this transmission route, such as face mask use, physical distancing, and disinfecting one's hands and environment [4], and now, in addition, advancing toward comprehensive vaccination. Societal measures including contact tracing and testing are also effective [5]. Symptom screening especially of fever is effective at identifying symptomatic patients and could be useful when the speed of disease spread outpaces contact tracing [6]. While symptom screening is limited by the fact that up to half of patients that test positive are asymptomatic [7,8], there is value in exploring all possible approaches to reduce disease spread.

To understand symptomatic patients early on in the pandemic, we previously investigated whether there is a likely order of symptom onset in respiratory diseases, including COVID-19, to discern any differences that could be beneficial for the public towards early recognition and course of a symptomatic infection. To determine the most and least likely order of symptoms of these diseases, we developed a mathematical model that is represented as a directed acyclic graph, where we perform a counting method on a Hasse diagram, which we used to visually represent a graded partially ordered set (poset). The graded poset of this model consists of an ordering of all possible combinations of symptoms that we are analyzing, where the top row with a single node indicates no symptoms, the next row indicates one specific symptom continuing until the bottom row consisting of a single node that indicates all symptoms have been experienced. Using this graphical model called the Stochastic Progression Model, we applied a Markov Process and implemented the model over different datasets that represent symptom frequency in a patient population to find most likely and least likely order of symptoms in these diseases [9]. Similar approaches that include Hasse Diagrams to represent posets for the purpose of modeling progressions as a Markov Process have been previously used to model the accumulation of mutations in cancer and Human Immunodeficiency Virus (HIV) [10,11]. Additionally, work in environmental science has included these elements of the theory of posets [12]. In our previous work, we used this model to distinguish individuals infected with

COVID-19 and influenza with acceptable recall and selectivity [9]. These results suggest that it might be possible to differentiate diseases by order of discernible symptoms. Discernible symptoms are objective and easy for patients to recognize and amenable to self-reporting. The model predicted that the most likely order of discernible symptoms in COVID-19 was first fever, then cough, followed by nausea and/or vomiting, and finally, diarrhea. This finding supported fever checking as an appropriate additional screening method as public spaces reopened [9]. Nonetheless, it is still unclear whether this order of symptom onset determined from the initial outbreak in China holds during the subsequent outbreak in the United States of America (USA).

Indeed, the pandemic has spread throughout the world and SARS-CoV-2 has mutated in multiple ways, and a large number of individuals with multiple risk factors have been diagnosed with COVID-19. A SARS-CoV-2 infection appears to affect patients differently due to various factors, and there is evidence of specific mutations affecting COVID-19 severity [13]. A different variant of SARS-CoV-2 had become dominant in the USA as of July 2020. The increasingly prevalent SARS-CoV-2 variant in North America and Europe during the timespan of the USA study was the D614G variant of the G clade, which did not progress in Asia until later [14]. The most signifying characteristic of this early variant is the mutation in the spike protein S-D614G [14,15]. This variant is responsible for the greatest number of cases in the pandemic, as of May 2020, and creates a higher upper respiratory tract viral load leading to a higher probability of transmission than the Wuhan reference strain, but it does not appear to induce a more severe form of disease [16]. Besides mutation, comorbidities of patients have been linked to changes in disease progression. The Centers for Disease Control and Prevention (CDC) found that COVID-19 patients that had underlying health conditions were 6 times more likely to be hospitalized and 12 times more likely to die [17]. Similarly, patients with chronic obstructive pulmonary disease (COPD), which is a condition that is associated with reduced airflow in the lungs, experience a higher likelihood of severity and death [18]. On the other hand, it has been suggested that individuals with hypertension or diabetes who are also infected with COVID-19 have been linked to worse outcomes [19]. We explore comorbidities including hypertension, diabetes, COPD, HIV, and cancer in this work. To what extent the likely order of symptoms vary with mutation and comorbidity could provide insight into how infectious diseases affect hosts.

In this study, we intended to implement our previous model with newly available data to ascertain whether the previously determined likely order of symptoms is consistent with other cases of COVID-19 that are characterized by a variant of SARS-CoV-2 and patient comorbidities. To this end, we first analyzed a large dataset from the USA, that consists of approximately 70% of D614G variant cases at this time, to compare to our previous results that were calculated from a large dataset from China, that consists of approximately 2% of D614G variant cases at the time of data collection [16]. We then analyzed other datasets to determine whether the D614G mutation is correlated with the predicted likely order of symptom onset. Finally, we tested whether the confounding factors, weather, patient age, and comorbidities, affect the predicted likely symptom order.

## Results

### Most likely initial symptom of COVID-19 varies between the initial outbreak in China and the subsequent spread to the USA

Early in the pandemic, we implemented the Stochastic Progression Model to model a large dataset from China to test whether the model predicts a specific symptom order [9], but whether this prediction of symptom order holds in other situations is unclear. We chose to

first perform the same analysis over a set of 373,883 cases in the USA that were collected from January 22 to May 30, 2020 [17] as we did previously for the initial dataset from China, which was collected from February 16 to 24, 2020 [20]. Using the previous China dataset and this USA dataset (**S1 Table** and **S1 Appendix**), we constructed a Hasse Diagram, which illustrates states (*i.e.* sets of symptoms) and transitions between states, and highlighted the most likely path of transitions with bolded color (**Fig 1A**). Surprisingly, the most likely symptom order differed between the initial outbreak in China and the subsequent spread to the USA. Specifically, our model predicted that in the USA dataset cough is more likely than fever to be the initial symptom and the inverse in the China dataset.

We next investigated the difference in the order of fever and cough as initial symptoms using transition probabilities. These are defined as the likelihood of a patient moving from one state to another and are calculated from the raw frequency data (**S1 Table** and **Fig 1B**) through the counting method on a Hasse Diagram [9]. We calculated the error of the transition probabilities of each dataset (**S1** and **S2 Figs**) to rank transitions that differ with a probability greater than the error towards determining the relative likelihood of paths. The methods used to calculate state and transition probabilities as well as the error used to determine paths are discussed in greater detail in the Materials and Methods section. In the patients during the initial outbreak in China, the model predicted the most likely initial symptom to be fever, with a transition probability of 0.769 (**Fig 1C**). In the patients during the later outbreak in the USA, the model predicted cough as most likely to occur first, with a transition probability of 0.475 (**Fig 1D**). In other words, our model predicts 76.9% of the patients in the dataset from China will experience fever first rather than the other three symptoms analyzed, whereas 47.5% of the patients in the dataset from the USA will experience cough first. The errors of the transition probabilities using the China and USA datasets are 0.010 (**S1 Fig**) and 0.063 (**S2 Fig**), respectively. The second most likely initial symptom from the China dataset is cough, with a transition probability of 0.221, while the second most likely initial symptom from the USA dataset is fever, with a transition probability of 0.353 (**Fig 1C and 1D**). Thus, our model predicted that fever was three times more likely than cough to occur first in the China dataset, while cough was only about one third more likely to occur first than fever in the USA dataset.

## Gastrointestinal symptom order varies between the initial outbreak in China and the subsequent spread to the USA

We modeled the first, second, and third most likely orders of symptom onset of COVID-19 for the same datasets used above, that occurred during the initial outbreak in China (**Fig 2A**) and during the subsequent outbreak in the USA (**Fig 2B**) to further observe any predicted variance in the order of symptoms of COVID-19. Besides fever or cough being the most likely initial symptom, another notable difference between these countries occurs in the first and second most likely paths. For the dataset from China, the third symptom in the first and second most likely paths is nausea/vomiting, whereas for the USA dataset the third symptom in the first and second most likely paths is diarrhea (**Fig 2C and 2D**). The transition probability of nausea/vomiting occurring third in both the first and second most likely paths is 0.575 for the dataset from China (**Fig 2C and 2D**). In contrast, the transition probability of diarrhea occurring as the third symptom is 0.647 in the first and second most likely paths as determined from the dataset from USA (**Fig 2C and 2D**). The error of the transition probabilities of the model of the China dataset is 0.010 and the error of the transition probabilities of the USA dataset is 0.063 (**S1** and **S2 Figs**). These predictions of likely paths suggest a difference in how SARS-CoV-2 affects the gastrointestinal (GI) tract: the upper GI tract more often is affected earlier in the cases from China (i.e., nausea/vomiting), and in the USA, the lower GI tract more often is

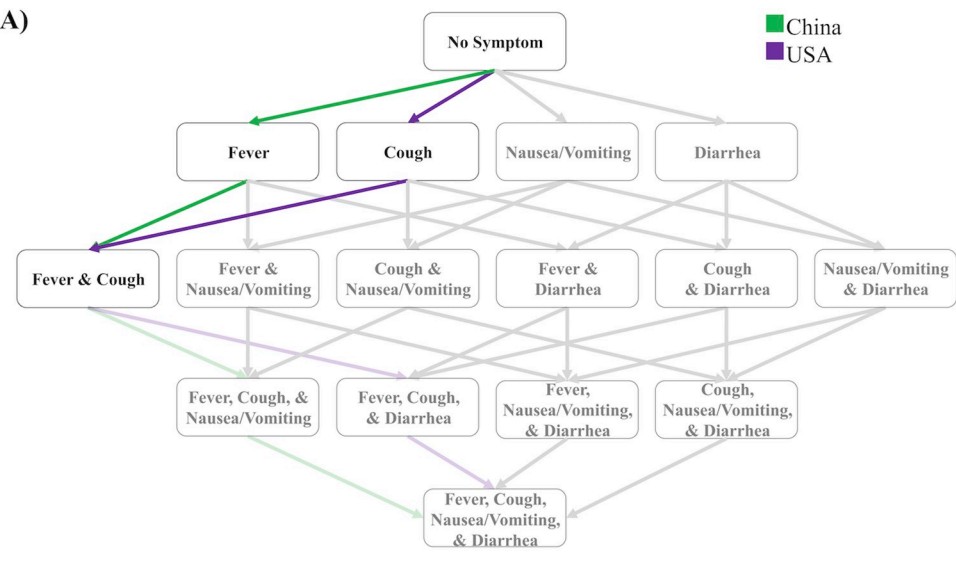

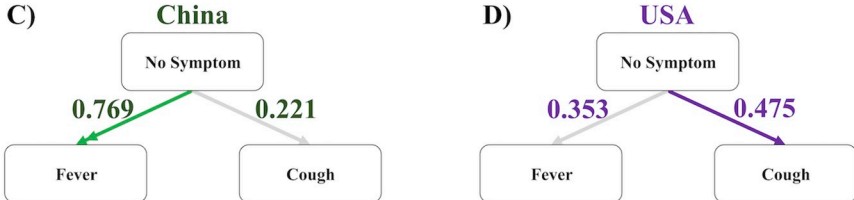

**Fig 1. Initial symptom differences between cases from China and from the USA.** A) A Hasse Diagram depicting the likeliest path of discernible symptom onset, highlighting the most likely initial symptoms, for China (green) and USA (purple). B) Table of raw frequency data specifying the number and the percentage of patients that experienced fever and cough in the China and USA datasets. C) Abridged Hasse Diagram depicting the transition probabilities from no symptom to fever or cough in the most likely path of discernible symptoms for COVID-19 patients in China. The double arrow signifies that the transition probability of the most likely first symptom is more than double the transition probability second most likely first symptom. The error of these probabilities is 0.010. D) Abridged Hasse Diagram depicting the transition probabilities from no symptom to fever or cough in the most likely path of discernible symptoms for COVID-19 patients in the USA. The error of these probabilities is 0.063.

affected earlier (i.e., diarrhea). In contrast, a key similarity is in the third most likely paths. The GI symptoms have transition probabilities that are too similar to state that one is likely to occur before the other, and therefore they are equally likely to occur as the first and second discernible symptom in these cases, and we group them as one node when observing the probabilities of the path (**Fig 2E**). Despite the similarity in the third most likely path order, nausea/vomiting and/or diarrhea are far less likely to be the first discernible symptom in the China cases during the initial outbreak (i.e., transition probability of 0.010) compared to the USA cases during the subsequent outbreak (i.e., transition probability of 0.172). We reasoned that if

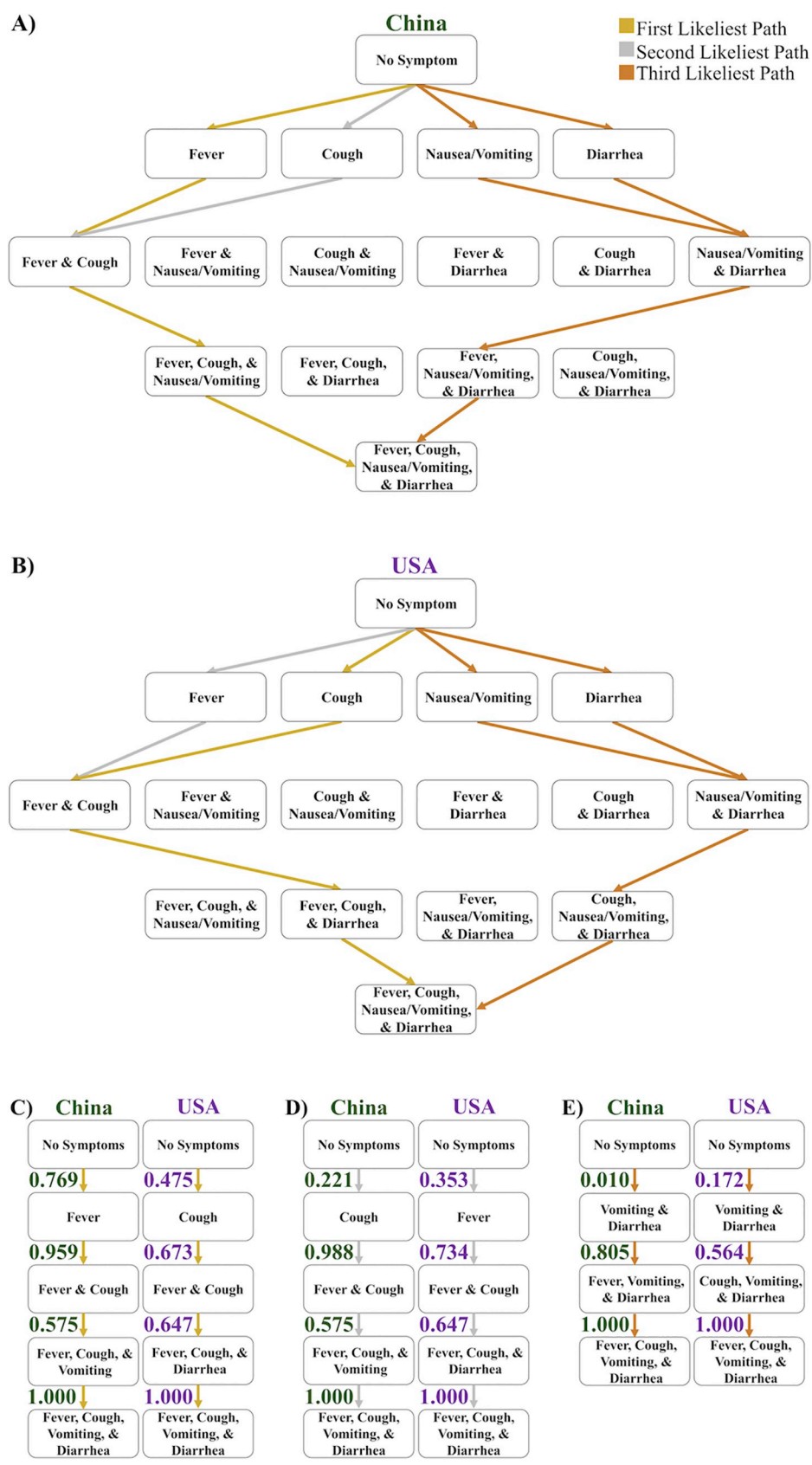

**Fig 2. The three likeliest paths of discernible symptom order in China and the USA.** A) A Hasse Diagram depicting the first (gold), second (silver), and third (bronze) likeliest paths of the order of discernible symptoms in China. In the case where the gold and silver lines converge, they both follow the same path. B) A Hasse Diagram depicting the first, second, and third likeliest paths of the order of discernible symptoms in USA. In the case where the gold and silver lines converge, they both follow the same path. C) The most likely path of discernible symptom order in China (green) and USA (purple). D) The second likeliest path of discernible symptom order in China and USA. E) The third likeliest path of discernible symptom order in China and USA. The errors of the probabilities of the paths using the datasets from China and USA dataset are 0.010 and 0.063, respectively.

diarrhea was a recurrent initial symptom, people in the USA would start to search for the term "diarrhea" more in March 2020 as the pandemic spread. Indeed, using Google Trends, we found a large relative increase in "diarrhea" keyword searches in the USA in March 2020 compared to the previous year (**S3 Fig**) [21]. After the first discernible symptom in each of the third most likely paths, the remaining discernible symptoms follow the order of the first most likely paths. Thus, in China the upper GI tract symptom nausea/vomiting was more likely to occur first, while in the USA the lower GI tract symptom diarrhea was more likely to occur first. Thus, based on these datasets, our model predicts that among GI symptoms, nausea/vomiting occurs before diarrhea in the cases from the initial outbreak in China and diarrhea is more likely to occur before nausea/vomiting in the subsequent cases in the USA.

## Symptom order correlates with variant of SARS-COV-2

Given the variations in symptom order observed between the early outbreak in China and later outbreak in the USA, we hypothesized that the symptom order is related to mutations in the virus. The Wuhan reference strain was dominant in Asia during the China study [16], but the D614G mutation was highly prevalent in North America during the USA study [22]. In order to further elucidate the relationship between symptom order and the D614G mutation, we implemented the Stochastic Progression Model as we did above to model two datasets from two other geographical regions with available symptom frequency data. To validate this observed difference in likely symptom order, we performed the analysis for datasets from Hong Kong and Brazil [23,24]. The time of data collection for the dataset from Hong Kong was during a period of the Wuhan reference strain dominating the region, whereas the D614G variant was predominant in Brazil at the time of that study [16]. The most likely path determined from Hong Kong was consistent with the dataset from China, except diarrhea is the third most likely symptom in Hong Kong as opposed to nausea/vomiting occurring third in China (**Figs 3A and S4A**) [20,23]. We calculated the transition probabilities from raw frequency data (**Fig 3B and S2 Table**). The error of the transition probabilities of the implementation of the Hong Kong dataset is 0.017 (**S5 Fig**). The most likely order of symptoms determined from the Brazil dataset is the same as the most likely order determined from the USA (**S4B Fig**) [17,24]. The error of the transition probabilities of the implementation of the Brazil dataset is 0.019 (**S6 Fig**). The raw frequency of fever and cough occurrence is provided to allow comparison to the calculated transition probabilities of the initial two symptoms in the implementation of the Hong Kong and the Brazil datasets (**Fig 3B–3D and S2 Table**). Similar to the nationwide results in China and the USA, respectively, our model predicted that the symptomatic patients in Hong Kong are very likely to first experience fever then cough (**Fig 3C**) while symptomatic patients from the Brazil study are only slightly more likely to first experience cough then fever (**Fig 3D**). The D614G mutation in SARS-CoV-2 has been linked to COVID-19 pathology, as it results in higher viral infectivity and antigenicity [25,26], so the mutation might also affect order and prevalence of symptoms. Therefore, these results support the notion that the likely order of symptoms in COVID-19 predicted by our model is linked to the D614G mutation in SARS-CoV-2.

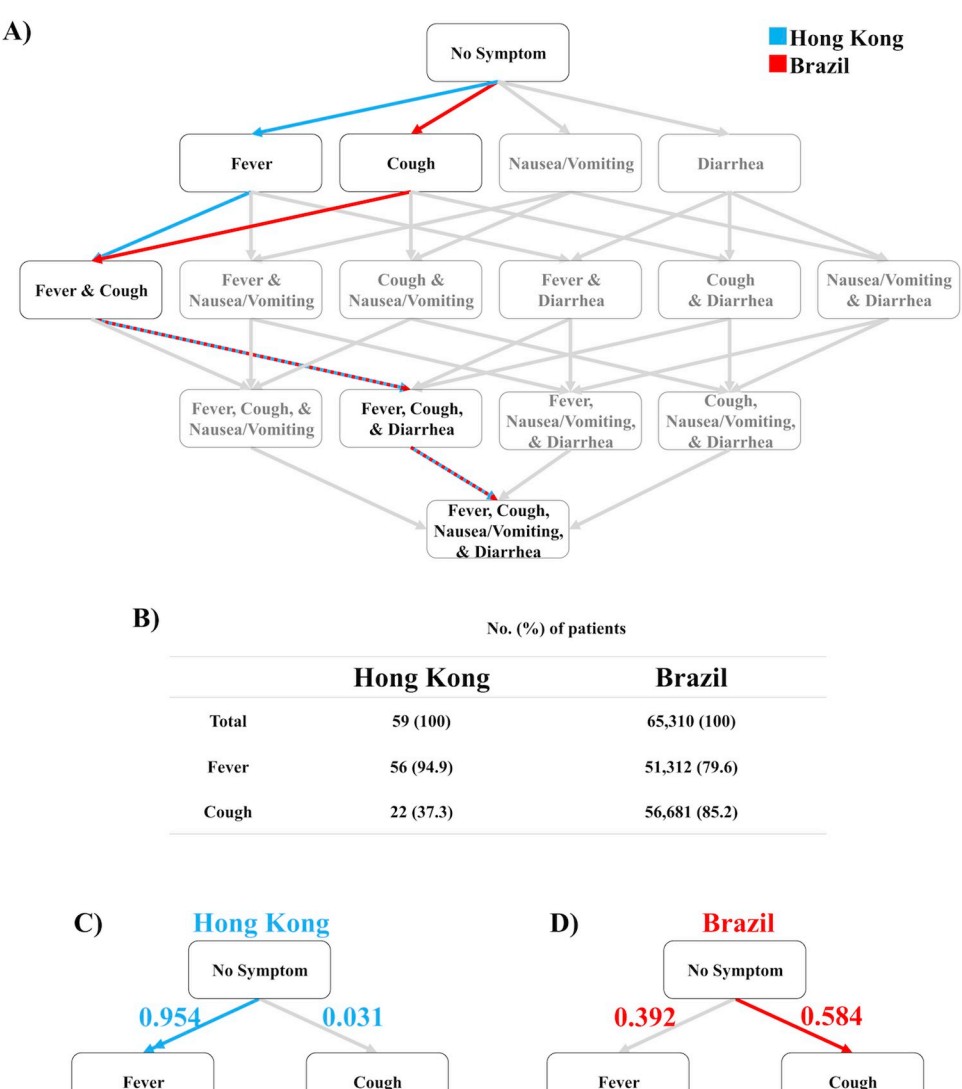

**Fig 3. The likeliest paths of discernible symptom order in Hong Kong and Brazil.** A) The most likely paths of the order of discernible symptoms of COVID-19 patients in Hong Kong (blue) and Brazil (red). Dashed red and blue lines denote transitions where both regions' most likely path is the same. B) Table of raw frequency data specifying the number and the percentage of patients that experienced fever and cough in the Hong Kong and Brazil datasets. (The study used for patients in Brazil reported the frequency of fever in 65,310 patients and the frequency of cough in 66,514 patients.) C) Abridged Hasse Diagram depicting the transition probabilities from no symptom to fever or cough in the most likely path of discernible symptoms for COVID-19 patients in Hong Kong. The double arrow signifies that the transition probability of the most likely first symptom is more than double the transition probability second most likely first symptom. The error of the transition probabilities is 0.017. D) Abridged Hasse Diagram depicting the transition probabilities from no symptom to fever or cough in the most likely path of discernible symptoms for COVID-19 patients in Brazil. The error of the transition probabilities is 0.019.

## Symptom order in Japan changes with introduction of D614G variant

If the hypothesis that symptom order depends on D614G mutation is correct, the order of symptoms should change in a population as the D614G mutation becomes more prominent. Uniquely, patients in Japan initially had the Wuhan reference strain until the D614G variant was first found there in early March [16]. From that point, about 9 out of 10 of cases in Japan were the mutated virus [16]. So, we implemented the Stochastic Progression Model once again

and created Hasse Diagrams to mathematically model a dataset from Japan from before the D614G mutation was prominent (**Fig 4A**) and a dataset from Japan that consisted of mostly cases after the D614G mutation emerged (**Fig 4B**). The early dataset did not report nausea or vomiting, so we did not include these symptoms in our analysis here [27]. We calculated the transition probabilities from raw frequency data (**Fig 4C and S3 Table**). In the early data set, fever occurred first by a high likelihood (**Fig 4D**) [27]. However, when we implemented the model on the later dataset, the results indicated an equal likelihood of cough or fever occurring first (**Fig 4E**) [28], as if the patient population was shifting from a China-like disease phenotype to a USA-like phenotype. Before the mutation, the transition probability of fever occurring first was 0.803 (**Fig 4D**). After the mutation, the probabilities of cough first and fever first are 0.480 and 0.464, respectively (**Fig 4E**). The error of the transition probabilities of the earlier dataset is 0.002 (**S7 Fig**). The error of the transition probabilities of the later dataset is 0.036 (**S8 Fig**). Because the error is greater than the difference between the initial cough and fever transition probabilities, fever and cough are equally likely to occur first. These transition probabilities are similar to those from China and the USA datasets. Furthermore, diarrhea became more prevalent with the mutation, as it was in the USA (**S9 Fig**). The transition probability of diarrhea occurring as the first symptom is 0.002 and 0.057 before and after the mutation, respectively. Thus, before the D614G mutation became prominent in Japan, the order of symptoms matched China. However, after the D614G mutation became prominent in Japan, the order of symptoms matched the USA. These results support the notion that the D614G variant induces a different symptom order in symptomatic patients.

We reasoned that patients in Japan would be searching for symptoms in accordance with this timeline. To test this, we used Google Trends to investigate whether people in Japan were searching more for fever, cough, or diarrhea in certain months in early 2020 [21]. Early in the pandemic in January and February 2020, there were not many cases in Japan, but there were relatively more searches of fever (**Fig 4F**), which matches the pathology of the disease in China (**Figs 1 and 2**) and Japan pre-mutation (**Fig 4A**). Additionally, in March when the D614G mutation became predominant in Japan, there was an increase of cough searches much greater than the increase of searches of fever that continued in April (**Fig 4F**). Also, in March, diarrhea searches increased (**Fig 4F**) as they had in the USA (**S3 Fig**). Furthermore, the magnitudes of these changes indicate that more people were searching these terms in March 2020 than the previous year, as the number of D614G patients increased throughout Japan. Thus, in Japan, the predicted order of symptoms tracked with the mutation spreading throughout the population.

## Weather, age, and comorbidities do not affect symptom order

Given the evidence that the D614G variant affects symptom order, we tested the effect of potential confounding factors on symptom order. Region does not seem to affect symptom order, as we have analyzed the symptom order in China, Hong Kong, and Japan for the Wuhan reference strain and in the USA and Brazil for the D614G variant (**Figs 1, 3, and 4**). Climate and season affect the pathology of certain pathologies, so we tested whether weather affects our model's predictions of COVID-19 symptom order. The China and USA datasets were collected over vast regions and different seasons, so we could not control for the effects of weather [17,20]. Similarly, the Hong Kong dataset was collected in winter, and the Brazil dataset was collected in summer and fall, and these two regions have different climates [23,24]. Lastly, the two Japanese datasets were collected during different seasons [27,28]. To investigate weather as a confounding factor, we determined the most likely orders of symptom onset for datasets from cities in Asia (Shanghai, China and Osaka, Japan) that were characterized similarly by the Wuhan reference strain and differently by weather. Then, we compared the most

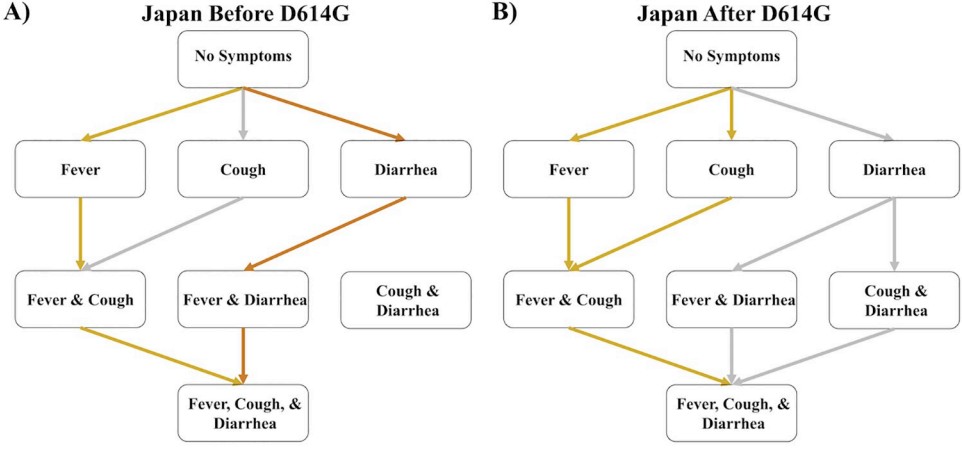

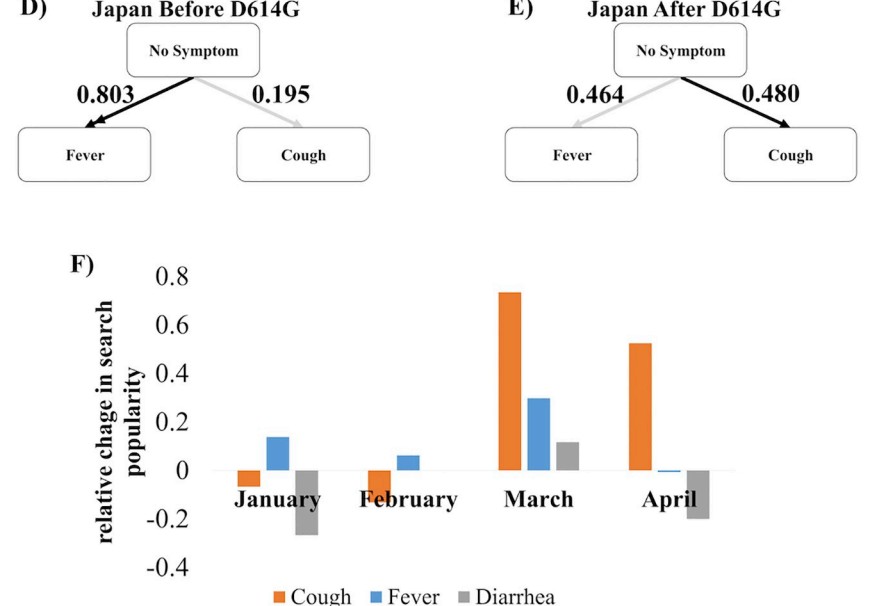

**Fig 4. The three likeliest paths of discernible symptom order in Japan before and after the D614G mutation became prominent.** A) The first (gold), second (silver), and third (bronze) likeliest paths of the order of discernible symptoms of COVID-19 patients before the D614G mutation became prominent. In the case where the gold and silver lines converge, they both follow the same path. B) The first, second, and third likeliest paths of the order of discernible symptoms of COVID-19 patients in Japan after the D614G mutation became prominent. C) Table of raw frequency data specifying the number and the percentage of patients that experienced fever and cough in the datasets of Japan before and after the D614G mutation became prominent. (The study used for symptom frequencies of COVID-19 patients after the outbreak of the D614G mutation reported symptoms of 2636 patients, except for cough, where only 2634 of the patients were recorded.) D) Abridged Hasse Diagram depicting the transition probabilities from no symptom to fever or cough before the D614G mutation became prominent. The double arrow signifies that the

transition probability of the most likely first symptom is more than double the transition probability second most likely first symptom. The error of the transition probabilities is 0.002. E) Abridged Hasse Diagram depicting the transition probabilities from no symptom to fever or cough after the D614G mutation became prominent. The error of the transition probabilities is 0.036. F) A bar graph displaying the relative change, compared to the previous year, of search popularity of the terms, "cough," "fever," and "diarrhea," in the months, January, February, March, and April 2020 calculated from Google Trends.

likely orders of symptom onsets with that of the large China dataset [20,27,29] and found them to be consistent (**Fig 5A**). The errors of the transition probabilities for Shanghai and Osaka were 0.004 and 0.002, respectively (**S7 and S10 Figs**) and were calculated from raw frequency data (**S4 Table**). We performed the same analysis for different cities in the USA (Detroit, Michigan, New York, New York, and Atlanta, Georgia) [30–32] and found the orders were consistent with the USA dataset (**Fig 5B**) [17]. The errors of the transition probabilities for Detroit, New York, and Atlanta were 0.049, 0.044, and 0.037, respectively (**S11–S13 Figs**). Thus, climate did not change the predicted symptom order.

Age affects the pathology of COVID-19. The datasets we analyzed report differing median and interquartile ranges (IQR) of the age of the patients, so we investigated the effect of age on likely symptom order using a dataset characterized by the Wuhan reference strain and one characterized by the D614G variant. The earlier dataset from Japan reported symptom data by

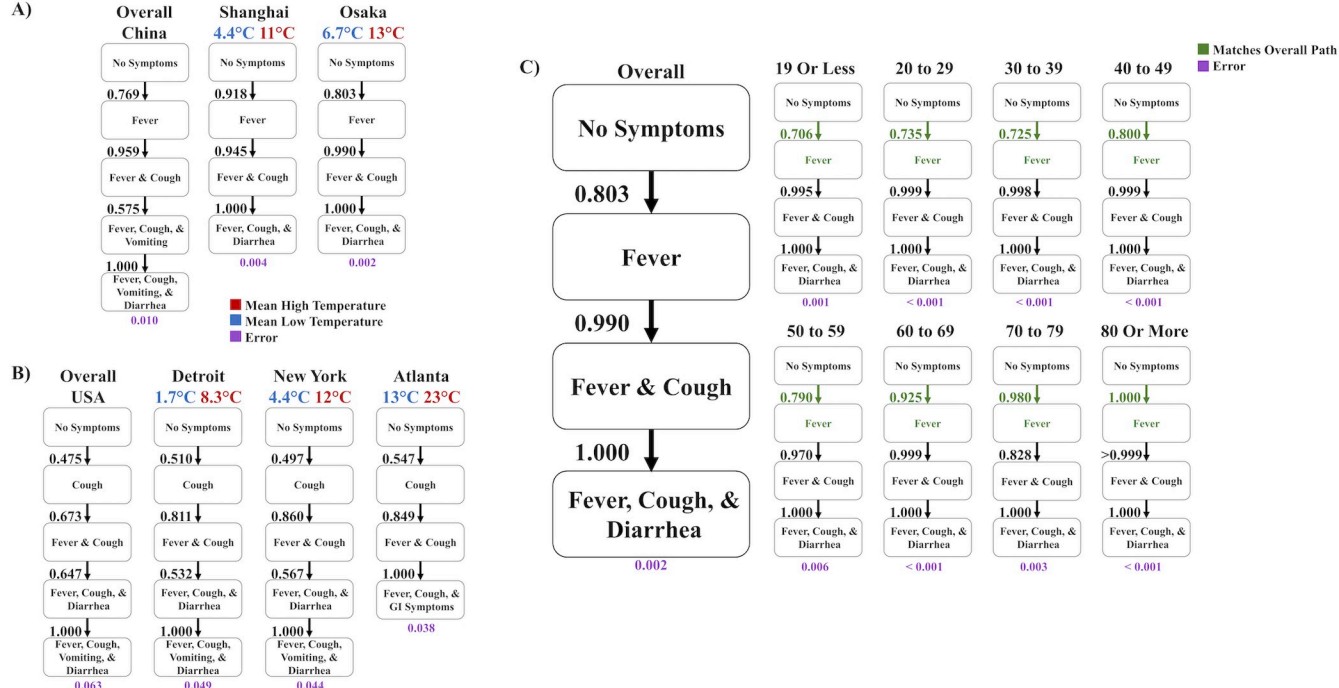

**Fig 5. The likeliest paths of discernible symptom order in cities with varying weather and in Japan with varying age.** A) The most likely path of discernible symptom order using the overall dataset from China is shown on the left. The middle and right columns are the most likely path of discernible symptom order of datasets in Shanghai, China and Osaka, Japan, when the Wuhan reference strain was prominent. The mean high temperature during the time that data was collected is written in red, the mean low temperature during the time that data was collected is written in blue, and the error of each implementation is written in purple. B) The most likely path of discernible symptom order using the overall dataset from USA is shown on the left. The other columns are the most likely path of discernible symptom order of datasets representing, from left to right, Detroit, New York, and Atlanta in the USA, when the D614G variant was prominent. The mean high temperature during the time that data was collected is written in red, the mean low temperature during the time that data was collected is written in blue, and the error of each implementation is written in purple. C) The most likely path of discernible symptom order for all patients in the Japan dataset (in the earlier time frame and characterized by the Wuhan reference strain) is shown on the left. All other columns are the most likely path of discernible symptom order by age groups from the overall set. The initial transition is written in green if it is consistent with the overall most likely path. The error of each implementation is written in purple.

age [27]. So, we determined the most likely order of symptom onset for each age group, which were consistent with the overall one (**Fig 5C**). The error of the transition probabilities is 0.002 (**S7 Fig**). The USA dataset also reported symptom order by various age groups, so we determined the most likely order of symptom onset for each age group in the USA dataset (**S14 Fig**) [17]. Here, the error of the transition probabilities is 0.063 (**S2 Fig**). All age groups follow the same most likely order of symptom onset as the overall, except the model predicted in the youngest group (19 years old or less) and the two oldest groups (70 to 79 years old and 80 years old or more) that indicate either fever or cough is first by an indiscernible difference. However, all groups' initial transition probabilities of fever and cough are similar (approximately 0.4 to 0.5), indicating that likely symptom order is consistent by age group in this dataset. Thus, for both the Wuhan reference strain (**Fig 5C**) and the D614G variant (**S14 Fig**), we found that the most likely order of symptom onset did not change by age group.

Because we observed that symptom order changes by viral variant between the initial outbreak in China and the subsequent outbreak in the USA and in Japan and studies link the D614G mutation with changes in disease pathology [16,25,26], we hypothesized that these changes in likely symptom order are not due to comorbidities of the patients. To this end, we modeled datasets containing comorbidities to compare to the model predictions using datasets from China and the USA. Making Hasse Diagrams using a dataset of COVID-19 patients with at least one comorbidity in China [33] and a separate dataset of COVID-19 patients of which 94% had at least one comorbidity in the USA [31] (**S5 Table**), we determined the most likely orders of symptoms did not change because of comorbidities (**S15 and S16 Figs**). The transition probabilities of the most likely path in cases from China are very similar between the entire dataset and those with comorbidities (**S17 Fig**). The error of the transition probabilities of the main dataset and comorbidity dataset from China is 0.010 and 0.013, respectively (**S1 and S18 Figs**). Similarly, when observing the most likely path in the USA datasets with comorbidities, the order of symptom onset was the same as the order determined from the main USA dataset (**S19 Fig**). The error of the transition probabilities of the main dataset and comorbidity dataset of the USA is 0.063 and 0.044, respectively (**S2 and S11 Figs**). In the USA datasets, we studied two additional comorbidities, namely, chronic obstructive pulmonary disease (COPD) and human immunodeficiency virus (HIV). Again, the model predicted that patients with these comorbidities would have the same most likely order of respiratory symptoms (**S20A–S20C Fig**), and the errors of the transition probabilities of the dataset containing patients with COPD and patients with HIV were 0.061 and 0.031, respectively (**S21 and S22 Figs**) [34,35]. Lastly, we investigated datasets including individuals diagnosed with cancer and COVID-19 in China and the USA and determined the most likely paths as well as displayed the raw frequency data for comparison (**S20A, S20D, and S20E Fig**) [36]. The only difference in these likely orders is that the third symptom is most likely to be diarrhea in cases with cancer (**S23A and S24A Figs**) as opposed to nausea/vomiting as it is in the overall dataset in China (**Fig 1**) [20,36], but we found that the most likely order of symptoms was the same in the dataset representing cases with cancer (**S23B and S24B Figs**) as the overall dataset in the USA (**Fig 1**) [17,37]. The error of the transition probabilities of this dataset containing cases with cancer in China and the USA is 0.026 and 0.030, respectively (**S25 and S26 Figs**). In these cases, cancer did not affect likely symptom order. Thus, these analyses of weather, age, and comorbidity indicated no change in predicted symptom order.

## Discussion

Our study highlights the complexity of how viral variant, weather, age, and host factors affect symptoms of infectious diseases. Here, we mathematically modeled datasets that include

clinical characteristics in China, the USA, Hong Kong, Brazil, and Japan to predict symptom order, as we had done previously with data from China [9]. The symptom order differed between the initial outbreak in China and the pandemic spread of the virus in the USA. The predicted most likely order of symptoms is cough, fever, diarrhea, and then nausea/vomiting in the USA during the time period, January 22 to May 30, 2020, while in China it is fever, cough, nausea/vomiting, and then diarrhea when using data from the initial outbreak, collected from February 16 to 24, 2020. Due to the prevalence of the D614G mutation in the USA, we hypothesized that this mutation is the cause of the variance between the results of China and the USA. To test this hypothesis, we analyzed datasets from Hong Kong and Brazil with high incidence of the Wuhan reference strain and the D614G variant, respectively. The results from this analysis of cases from Hong Kong and Brazil were consistent with our model's predictions of the first two most likely symptoms. To further test our hypothesis, we analyzed datasets in Japan where the dominant strain changed from the Wuhan reference strain to D614G variant. COVID-19 cases in Japan were characterized as the Wuhan reference strain in January and February 2020, but after the first emergence of the virus with the D614G mutation in early March, approximately 90% of COVID-19 cases were caused by the mutated virus. We performed the analysis again on a Japan dataset that represents data when the Wuhan reference strain was prominent and a later Japan dataset when the D614G variant was prominent, and again our model's predictions were consistent with our hypothesis that symptom order depends on mutation. We also performed an analysis of Google Trends of discernible symptom search terms over early months of 2020, which supported our model's predicted symptom orders in various countries. Nonetheless, we cannot claim to what extent these trends we observed were related to increased prevalence of COVID-19 in the media and how the coverage varied between countries. Finally, we considered other factors that we hypothesized may affect likely symptom order. Here, we analyzed the most likely order of symptom onset by age and seasonal or climate changes, and we found that viral variant has a stronger effect on symptom order than age and season. Our study also included datasets from varying countries for both the Wuhan reference strain and the D614G variant. The results from those analyses indicate no change in symptom order due to country. We then analyzed datasets from China and the USA of patients with COVID-19 and comorbidities and found that the predicted order did not change when comparing results from patients with and without comorbidities. Thus, our model predicts that the order of symptom onset in COVID-19 patients are most dependent on the D614G mutation and does not change with comorbidities, age, or weather/climate. While others have proposed that SARS-CoV-2 mutation affects olfactory and gustatory symptoms [38], diarrhea [39], and COVID-19 pathology [25,26], our results suggest that symptom order may be affected by the D614G variant.

Comorbidities may nonetheless affect symptom order in other ways. Our model predicts differences in the order of GI symptoms between cancer and non-cancer patients with COVID-19 in China. Also, other studies report inconsistent patterns of symptom occurrence in COVID-19 patients with kidney diseases. In China, where fever is the most common initial symptom generally, incidence of fever and cough seems to be reduced in a small study of hemodialysis patients [40], while a larger study of hemodialysis reported a high frequency of fever in comparison to cough [41]. In contrast, a study in the USA found that each kidney transplant recipient with COVID-19 experienced fever, which was by far the most common symptom in this group [42]. This observation in kidney transplant patients contrasts with our study in the general population and other comorbidities, as we found that the most likely first symptom in the USA is cough. Besides analysis of various comorbidities, further work is needed to prove that the D614G mutation causes changes in discernible symptom order, as our current analysis is associative. We confirmed the change in order of fever and cough, but

we were unable to compare GI symptoms due to the lack of reporting in the Japanese datasets. It is possible that if nausea/vomiting was reported, it could have been the most likely first or second symptom, but because both fever and cough occur prior to GI symptoms with a high likelihood in all other datasets, it is unlikely. The results of our model presented here suggests that there is an order of symptom onset and this order changes with the D614G mutation, but how the variant affects the order of GI symptoms among themselves and relative to respiratory symptoms requires further validation, which requires additional data. As both smartphone applications developed by researchers [43,44] and clinicians themselves record and report the order of symptoms in COVID-19 patients [45], the conclusions of our model can be validated prospectively with patient data. Symptom order data from patients in countries with widespread testing available would indicate whether there is a symptom order of COVID-19 different from other respiratory diseases so that symptom order may be studied more in depth [43,44]. If a symptom order is verified in countries with widespread testing, data in countries without widespread access to testing would then determine whether the order holds there.

Our analysis investigates patterns in symptom order in different patient groups and the causes of these changes. It does not predict the symptom order of all patients. To investigate these patterns, we modeled four common and easily discernible symptoms, which we analyzed using our model previously [9]. In contrast, vague and subjective symptoms are prone to human error due to self-reporting. Also, rare symptoms do not perform well in this model, because the model may incorrectly predict that rare symptoms occur later. For example, loss of smell and taste has emerged as a telltale indicator of COVID-19 [46–48], and there is evidence that it is often an early symptom [43,49,50], but this symptom occurs in a small fraction of patients [43,48] and is self-reported by more than 20% of patients that eventually test negative [43]. Due to these reasons, we analyzed changes in likely order of discernable symptom onset in this study.

Currently, there is no formal metric describing the confidence associated with each specific path of likely symptoms, but in our previous application of the Stochastic Progression Model to likely symptom order of COVID-19, we were able to distinguish COVID-19 and influenza patients by symptom order with acceptable recall and selectivity [9]. Until we develop an appropriate confidence interval for the Stochastic Progression Model, we use a conservative error estimate based on state and transition probabilities to differentiate likely paths within each implementation of the model. As a result, we cannot explicitly state the confidence of each likely path. We are not proposing that the model predicts the likelihood of symptom order. Instead, we note that there are distinct features of symptom progression by datasets that correlate with the predominate viral variant.

This work highlights the need for new forms of data to be collected to develop strategies to study pandemics and symptomatic patient experience. As others have called for mobility data to enhance modeling of disease spread [51], widespread availability of ordered symptom data will enable the testing of the hypotheses generated here. Also, the time between symptoms and the time after infection or diagnosis that a symptom occurs could be investigated. Other researchers have studied symptom progression using prospective data collected in real-time and found that fever and cough occur early after infection with similar likelihood in the USA and United Kingdom [43,52]. These data are consistent with our findings, as these countries have a predominance of the D614G variant [16]. Similarly, a clinical study performed in China found that the most common first symptom in COVID-19 patients was fever, and this was unaffected by comorbidity [53]. Along with these studies, our current study underscores the potential value of symptom order towards understanding disease progression, especially in cases including mutating viral strains. Interestingly, the D614G variant has been shown to have a higher probability of transmission, but not a more severe disease [16], and here our

data suggests that symptom order changes as well. Thus, we propose the hypothesis that cough occurring earlier with the D614G variant could contribute to its higher transmissibility [54], because symptomatic people cough thereby spreading the virus before they are incapacitated with fever. This hypothesis is consistent with a report that suggest the D614G variant causes a higher upper respiratory tract viral load in patients [16], which implies that the patients might need to cough more and there is more virus spread per cough. If true, the widespread use of masks could be more important where the D614G and other similar variants predominate. Strategies to discern key differences in variants could be aided by the observation of symptom order. Such strategies will become increasingly important during the ongoing COVID-19 pandemic and future outbreaks of emerging variants of concern, which exhibit increased transmissibility (e.g., the WHO-labeled variants Alpha, Beta, and Gamma), increased virulence (e.g., the WHO-labeled variants Alpha and Gamma), and decreased antigenicity (e.g., the WHO-labeled Beta, Gamma, Epsilon, and Delta) [16,55–62].

## Materials & methods

### Data collection to study viral variant

We obtained multiple reports of COVID-19 symptomatic and asymptomatic cases around the world, such as the USA, China, and Japan, that contained symptom frequency data of patients. The reports cited here were used to simulate symptom data of individual patients. Then, we implemented the Stochastic Progression Model to model these various simulated datasets. The statistics and demographics of all the datasets that we described here is also tabulated in the S1 Appendix.

A large dataset consisting of 55,924 laboratory confirmed cases, which includes real-time polymerase chain reaction (RT-PCR) testing SARS-CoV-2, of COVID-19 in China was used. The patients in the dataset had a median age of 51 years (interquartile range [IQR] of 39 to 63 with minimum age of 2 days to maximum age of 100 and 77.8% being between 30 to 69), 23% of patients were not from Hubei, China, and 51.1% were male. The COVID-19 symptom data for this study included fever, dry cough, fatigue, sputum production, shortness of breath, sore throat, headache, myalgia or arthralgia, chills, nausea or vomiting, nasal congestion, diarrhea, and hemoptysis. This report found that these symptoms did not develop in symptomatic individuals on average for 5 to 6 days after infection. This data was collected from February 16 to 24, 2020 and organized by the World Health Organization [20]. The main dataset representing a large collection of COVID-19 infected individuals and their symptom data was reported by the CDC. This report contains 1,320,488 confirmed cases of COVID-19, with 373,883 of those cases recorded with known symptoms. We used this subset of 373,883 cases in this study. The data was collected from January 22 to May 30, 2020. The reported COVID-19 symptoms collected for this report were fever, cough, shortness of breath, myalgia, runny nose, sore throat, headache, nausea/vomiting, abdominal pain, diarrhea, and loss of smell or taste. The median age of confirmed cases in this report was 48 years (IQR of 33 to 63 and 64.3% between 30 to 69), 21.8% of patients had an underlying condition, and 48.9% were male [17].

Two validation datasets were collected to determine the order of symptoms of the Wuhan reference strain and the D614G variant in Hong Kong [23] and Brazil [24], respectively. These two datasets are independent from all other datasets collected for this study, because they represent cases from geographical regions that do not overlap with the regions from any other datasets. The study characterized by the Wuhan reference strain included 59 patients from Hong Kong who were diagnosed with virologically confirmed COVID-19 within February 2 to 29, 2020. The median age of these cases was 59 years (IQR of 44 to 68) and 46% were male. The symptoms reported in these patients from Hong Kong included fever, cough, dyspnea,

vomiting, diarrhea, and abdominal pain/discomfort [23]. The independent study used to confirm the order of discernible symptoms in cases of the D614G variant occurred in Brazil and included up to 67,180 individuals with COVID-19. These cases were confirmed by molecular diagnostic and clinical epidemiological criteria, are stored in the SIVEP-Gripe system, and occurred from March 1 to May 31, 2020. The median age of these confirmed cases was 59 years (IQR of 44 to 72), 83.7% had at least one comorbidity, and 58% were male. The symptoms reported by these patients in Brazil included fever, cough, dyspnea, respiratory discomfort, oxygen saturation <95%, sore throat, diarrhea, and vomiting [24].

Two different studies from Japan were used to determine if there were any changes in the order of symptom onset of COVID-19 due to a mutation of the virus. One study was a retrospective observational study, which was performed in the Osaka Prefecture of Japan and outlined 244 COVID-19 patients who were diagnosed with a positive PCR test from February 1 to March 31 in 2020. The largest age group was 40–49 years old (25.8%) followed by 20–29 (19.3%), then 50–59 (16%), and 51.2% were male. The most common symptoms in the group of symptomatic COVID-19 patients were fever, cough, headache, sore throat, general malaise, mucus/nasal obstruction/sneeze, low back pain, arthralgia, diarrhea, shortness of breath, and pneumonia [27]. Another observational study, containing 2,636 cases with symptom data (except cough was only recorded for 2,634 cases), ended on July 7, 2020 and collected data from 227 healthcare facilities throughout Japan. This study, from the COVIREGI-JP registry, followed an inclusion criterion of a positive SARS-CoV-2 test and being in an inpatient treatment care facility. This study does also point out that if one patient had a history of more than one COVID-19 hospitalizations and did meet the inclusion criteria mentioned prior, then each admission was included separately in the registry. The median age of this cohort was 56 (IQR of 40 to 71), 46.8% had a comorbidity, the median BMI of 2,024 participants was 23.1 (IQR of 20.5 to 26), and 58.9% were male. COVID-19 symptoms reported on admission to the hospital included fever, cough, productive cough, bloody sputum, fatigue, shortness of breath, diarrhea, sore throat, headache, dysgeusia, olfactory dysfunction, runny nose, confusion, arthralgia/myalgia, vomit, wheezing, chest pain, abdominal pain, seizure, and conjunctivitis with median duration of symptoms before admission was 7 days (IQR of 4 to 10 days) [28].

A dataset from Atlanta, Georgia was used to examine changes in symptom order due to changes in season and weather. This study consisted of 561 cases that were collected in 6 acute care hospitals and associated outpatient clinics in Atlanta and were confirmed by RT-PCR testing. The median age of patients was between 45 and 64 years old, 40% were obese, and 46% were male. The COVID-19 symptoms that were reported consisted of fever, cough, shortness of breath, headache, chills, arthralgia, myalgia, sore throat, loss of smell, loss of taste, and gastrointestinal symptoms [32]. The temperatures were found on WolframAlpha [63]. Another study was used to study changes in symptom order, and it was performed at the Shanghai Public Health Clinical Center in Shanghai, China. In this study, 249 COVID-19 patients were recruited from January 20 to February 6, 2020. They were diagnosed to have COVID-19 by detecting SARS-CoV-2 through RT-PCR on throat swabs. Seven of those patients were asymptomatic. The median age of these confirmed cases was 51 years (IQR of 36 to 64), 36.1% had at least one coexisting chronic condition, and 50.6% were male. The recorded symptoms of these COVID-19 patients at admission were fever, cough, fatigue, dizziness and headache, rhinorrhea, sore throat, diarrhea, and inappetence with the median duration of fever in all patients being 10 days [29]. Then, a study that encompasses 393 cases of COVID-19 in New York City, that were confirmed by RT-PCR performed on nasopharyngeal swab at a Manhattan Hospital from March 3 to 27, 2020 was also used. The median age of these confirmed cases was 62.2 years (IQR of 48.6 to 73.7), 25.2% had diabetes, 50.1% had hypertension, 5.1% had COPD, 12.5% had asthma, 13.7% had coronary artery disease, and 136 of 380 are obese, 62.6% were

not Caucasian, and 60.6% were male. The most common presenting symptoms in the symptomatic patients were fever, cough, diarrhea, nausea or vomiting, myalgia, and dyspnea [30].

## Data collection to study comorbidities

A study that took place at the Henry Ford Health System in Detroit, Michigan in USA from March 9 to 27, 2020 was used in the analysis of symptom order and comorbidities. The study consists of 463 laboratory confirmed cases of SARS-CoV-2 by the CDC interim guidelines for collecting, handling, and testing clinical specimens from persons under investigation, published February 14, 2020 [64]. The mean age of these cases was 57.5 years (standard deviation [SD] of 16.8), 94% had at least one confirmed comorbidity, the mean BMI was 33.6 (SD of 8.7), 27.9% were not African American, and 44.1% were male. The most common comorbidities recorded in this study were hypertension, obesity, chronic kidney disease, and diabetes. The COVID-19 symptoms at presentation in symptomatic patients in this study were cough, nasal congestion, dyspnea, fever, headache, myalgia, anorexia, nausea, vomiting, and diarrhea [31]. We then found comparable data from a retrospective case study, under the coordination of the National Health Commission, for 1,590 patients in China who were diagnosed with COVID-19 using high throughput sequencing or real-time, reverse-transcription PCR assay from nasal and pharyngeal swab specimens between December 11, 2019 and January 31, 2020. There is a subset of 399 patients in this dataset who had at least one comorbidity, which we used in our analysis. This subset had a mean age of 60.8 years (SD of 13.4) and 59.8% of them were male. The COVID-19 symptoms surveyed in this study were fever, conjunctival congestion, nasal congestion, headache, dry cough, pharyngodynia, productive cough, fatigue, hemoptysis, shortness of breath, nausea/vomiting, diarrhea, myalgia/arthralgia, and chill. The distribution of the patients' comorbidities recorded in this study is 67.4% had hypertension, 32.6% had diabetes, 14.8% had cardiovascular disease, 7.5% had cerebrovascular disease, 7% had hepatitis B infection, 6.0% had COPD, 5.3% had chronic kidney disease, 4.5% had malignancy, and 0.8% had immunodeficiency [33]. Additionally, a dataset to investigate order of symptom onset in individuals with COPD and COVID-19 was used. An in-depth analysis of 1,319 individuals, who were confirmed to have COPD included in their medical chart in the electronic health record (EHR) which requires physician confirmation, was performed by researchers at the Respiratory Institute and the Lerner Research Institute at the Cleveland Clinic in Ohio from March 8 to May 13, 2020. These patients had an average age of 67.7 years (SD of 12.1), 30.4% were not Caucasian, they had a mean BMI of 29.9 (SD of 9.0), and 45.7% were male. A subgroup of 164 patients also had laboratory confirmed COVID-19, confirmed by laboratory testing using the CDC RT-PCR SARS-CoV-2 assay, and this subgroup was used for modeling in this study. The reported COVID-19 symptoms by patients recorded in this study were cough, sputum production, dyspnea, fever, fatigue, flu like symptoms, loss of appetite, diarrhea, and vomiting. Some of these patients with COVID-19 and COPD have other comorbidities, such as diabetes (40.4%), hypertension (76.9%), coronary artery disease (37.5%), heart failure (34.4%), currently with cancer (7.3%), in remission from cancer (15.3%), immunosuppressive therapy (26.5%), inflammatory bowel disease (6.5%), smokes a pack a year (32.9%), and is a current smoker (32.2%) [35]. Specifically, to analyze the effects of the HIV on the symptom order, a retrospective study that included 93 individuals diagnosed with COVID-19, through a positive nucleic acid amplification test for SARS-CoV-2, from March 2 to April 15, 2020 in New York City who have also been diagnosed with HIV on their electronic medical record was applied to the model. This study was performed at the Icahn School of Medicine at Mount Sinai in New York, New York, and the data from five New York City emergency departments was collected.

The individuals included had a median age of 58 years (IQR of 52 to 65), 77.4% not Caucasian, a median BMI of 26.7 (IQR of 23.8 to 29.6), and 72% male. The COVID-19 symptoms present in the initial emergency department note and admission note recorded in this study were fever, altered mental status, congestion, sore throat, cough, shortness of breath, myalgia, anosmia, diarrhea, and headache. The distribution of comorbidities among these patients was 4.3% had autoimmune disease, 8.6% had cancer, 34.4% had diabetes, 18.3% had heart disease/coronary artery disease/congestive heart failure, 52.7% had hypertension, 26.9% had lung disease/asthma/COPD, 17.2% had chronic kidney disease, 7.5% had end-stage renal disease, and 5.4% had solid organ transplant [34].

In order to investigate the effects of cancer on symptom order in patients with COVID-19, a retrospective, multicenter, cohort study that consisted of 205 individuals who were laboratory confirmed using RT-PCR and next-generation sequencing analysis of nasopharyngeal swabs to be infected with SARS-CoV-2 as well as pathologically diagnosed with a malignant tumor was used (patients clinically diagnosed with COVID-19 were excluded and patients with a pathological diagnosis of a benign tumor were excluded). This study was led by the Wuhan Union Hospital, and the participating hospitals were the Cancer Center of Wuhan Union Hospital, West Branch of Wuhan Union Hospital, Jin Yin-tan Hospital, Wuhan Red Cross Hospital, the Central Hospital of Wuhan, Huanggang Central Hospital, the First People's Hospital Affiliated to Yangtze University, Xianning Central Hospital, and Suizhou Central Hospital. There was an enrollment window of January 13 to March 18, 2020. The patients in this retrospective study had a median age of 63 years (IQR of 56 to 70), 52% had a comorbidity besides cancer, and 47% were male. The symptoms that were recorded in this study were fever, chills, cough, sputum, chest congestion, shortness of breath, dyspnea, nausea or vomiting, diarrhea, and fatigue [36]. Similarly, a study performed in New York State contains 423 COVID-19 cases, which were confirmed by detecting SARS-CoV-2 RNA from a nasopharyngeal swab sample using the CDC protocol. These patients were also diagnosed with cancer on their electronic medical records. The data was collected from March 10 to April 7, 2020. Symptoms assessed at the time of COVID-19 testing were fever, shortness of breath, cough, and diarrhea. The distribution of the types of cancers that patients were diagnosed with is leukemia (8%), lymphoma (11%), myeloma (5%), breast (20%), colorectal (9%), lung (8%), prostate (6%), and others (32%), with 56% having metastatic disease. The age distribution of these patients was reported as 7 patients less than 18 years, 11 patients between 18 to 29 years, 19 patients between 30 to 39 years, 51 patients between 40 to 49 years, 101 patients between 50 to 59 years, 134 patients between 60 to 69 years, and 100 greater than 70 years, and 38% of patients were not Caucasian and 50% were male. Some of these patients in this study have other comorbidities. The distribution is 10% with asthma, 7% with COPD, 20% with diabetes, 20% with cardiac dysfunction, 9% with chronic kidney disease, 51% with hypertension, 45% with chemotherapy, 16% with chronic corticosteroid, 2% with chronic lymphopenia, and 7% with immune checkpoint inhibitors as well as 7% having a major surgery in the last 30 days. The distribution of the patients' BMI is that 13 were underweight (less than 18.5), 135 were normal (18.5 to 24.9), 146 were overweight (25.0 to 29.9), 109 were obese (30.0 to 39.9), and 20 were severely obese (greater than or equal to 40) [37].

## Implementation of the Stochastic Progression Model

The Stochastic Progression Model is described in detail in our previous publication, in which we used this model to find likely order of discernible symptoms in patients experiencing COVID-19 early on in the pandemic [9]. It was constructed in R (version 3.5.2) [65]. Briefly,

the Stochastic Progression Model is a directed acyclic graph that consists of nodes that represents a specific combination of symptoms that an individual has experienced up until this point and edges that represent an individual experiencing an additional new symptom, depicted as a Hasse Diagram on a graded poset. The resulting Hasse Diagrams were visualized using the hasse function in the hasseDiagram_0.1.3 library in R [66].

To model the progression of symptoms from static patient data, we first use the frequency of each symptom occurrence to simulate data for 500,000 theoretical patients. The raw discernible symptom frequency data used can be found in **S1**–**S5 Tables**. The results describe individuals that may have experienced any number of the symptoms studied. Here, we assumed that the onset of symptoms is independent and random. We then used these simulated datasets to construct the corresponding Hasse Diagrams. Again, each node represents a possible state of an individual by the specific symptoms that they have experienced, and edges represent the transition of a patient from one state to another that contains one more symptom. First, we assumed that each symptom occurs one at a time, even if there is only an extremely small amount of time between them. Second, we assumed that all patients would eventually experience all symptoms. Using this representation of nodes and edges and these assumptions, we created directed acyclic graphs in the form of Hasse Diagrams of graded posets and applied a Markov Process.

Next, we derived the state probabilities of the nodes and transition probabilities of the edges in the diagrams. State probabilities of nodes and transition probabilities of edges were calculated using symptom frequency data and are used to perform a counting method upon our model. The state probability of a node is calculated from dividing the total number of patients that experience a specific set of symptoms by the total number of patients that experience any combination of the same number of symptoms. The transition probabilities in the model are defined as the likelihood of moving from one state to another. As a result, the sum of all transition probabilities of edges with the same originating node is equal to one. The transition probability of an edge is calculated by dividing the total number of patients considered at the terminating node by the number of patients at any node that represents the same number of symptoms as the terminating node including the number of patients at that terminating node. Likelihoods of paths are found by applying a Markov Process along a path starting from the node with a set of all zeroes to the node ending with a set of all ones using each edge's respective transition probability. The error of a specific node was then calculated by a series of steps. First, each path to that node was determined. The transition probabilities of each edge in a path were multiplied to a single product. The product of each path leading to the node together are summed. Finally, that summed value is subtracted from the calculated state probability. The absolute value of that result is the error of that node. The value of the node with the highest error was then determined to be the error of that implementation of the model to serve as a conservative error estimate (**S1**, **S2**, **S5**–**S8**, **S10**–**S13**, **S18**, **S21**, **S22**, **S25**, **and S26 Figs**). The error of each implementation allows us to compare each edge within the same implementation. We finally determine which path is more likely by directly comparing transition probabilities of edges. If an edge from a node has a higher transition probability than any other edge from that node by a margin greater than the error, we identify that edge as more likely than any of the others from that node. The code can be viewed at https://github.com/j-larsen/Stochastic_Progression_of_COVID-19_Symptoms.

## Estimating prevalence of viral variant within populations

We estimated the percentage of the D614G variant and the Wuhan reference strain in our datasets using the results of a study that observed the changes in prevalence of these strains

globally [16]. The dataset collected early in the pandemic in China, from February 16 to 24, 2020, represents cases that occurred at a time when over 98% of cases were consisting of the Wuhan reference strain. The dataset collected during the subsequent outbreak in the USA, from January 22 to May 30, 2020, represents cases that occurred at a time when approximately 70% of cases were characterized by the D614G variant. Specifically, approximately 100% of cases in Hong Kong from February 2 to 29, 2020 were characterized by the Wuhan reference strain, and approximately 85% of cases in Brazil from March 1 to May 31, 2020 were characterized by the D614G variant. So, we modeled datasets from these cities in these time periods that consisted of cases that were almost exclusively of a specific type. The results were consistent with the earlier findings based on the representative countries. Lastly, we studied cases that occurred in Japan, because the initial outbreak in Japan was exclusively characterized by the Wuhan reference strain until late February 2020 and a separate outbreak occurred starting early March 2020 during which approximately 95% of cases were characterized by the D614G variant [16].

### Analysis of Google trends by country

We downloaded a metric that represents the popularity of searches per week of various symptoms as search terms in the USA and Japan during the time period from December 2018 to May 2020 [21]. The relative change in search popularity of a term of a specific month was calculated by subtracting the total search popularity for that term in the month from the total search popularity in the same month of the previous year, and then divide the result by the total search popularity in the same month of the previous year.

### Supporting information

**S1 Table. Raw frequency data specifying the number and percentage of COVID-19 patients that experienced discernible symptoms from clinical datasets from China [20] and USA [17].** These frequencies were used to simulate patients to find the likeliest paths of symptom onset for discernible symptoms of COVID-19. The dataset from China contains 55,924 patients, and the dataset from USA contains 373,883 patients.
(XLSX)

**S2 Table. Raw frequency data specifying the number and percentage of COVID-19 patients that experienced discernible symptoms from clinical datasets from Hong Kong [23] and Brazil [24].** These frequencies were used to simulate patients to find the likeliest paths of symptom onset for discernible symptoms of COVID-19. The dataset from Hong Kong contains 59 patients, and the dataset from Brazil contains at least 50,000 patients.
(XLSX)

**S3 Table. Raw frequency data specifying the number and percentage of COVID-19 patients that experienced discernible symptoms from clinical datasets in Japan before [27] and after [28] D614G mutation.** These frequencies were used to simulate patients to find the likeliest paths of symptom onset for discernible symptoms of COVID-19. The dataset from Japan before the outbreak of the D614G variant contains 244 patients, and the dataset from Japan after the outbreak of the D614G variant reports symptoms of 2,636 patients, except for cough, where only 2,634 of the patients were recorded.
(XLSX)

**S4 Table. Raw frequency data specifying the number and percentage of COVID-19 patients that experienced discernible symptoms from clinical datasets in Shanghai [29], Osaka [27], New York [30], and Atlanta [32].** These frequencies were used to simulate patients to find the likeliest paths of symptom onset for discernible symptoms of COVID-19.

The dataset from Shanghai, Osaka, Atlanta, and New York contains 249, 244, 531, and 393 patients, respectively.
(XLSX)

**S5 Table. Frequencies of COVID-19 discernible symptoms from clinical datasets in patients with COVID-19 and comorbidities.** The columns of the frequencies correspond to dataset. From left to right, they represent a dataset containing patients with COVID-19 and comorbidities in China [33], patients with COVID-19 and comorbidities in the USA [31], patients with COVID-19 and cancer in China [36], patients with COVID-19 and cancer in the USA [37], patients with COVID-19 and COPD in the USA [35], and patients with COVID-19 and HIV in the USA [34]. These frequencies were used to simulate patients to find the likeliest paths of symptom onset for discernible symptoms of COVID-19. The dataset representing patients with comorbidities from China and the USA contains 399 and 463, respectively. The dataset representing patients with cancer from China and the USA contains 205 and 423, respectively. The dataset representing patients with COPD and HIV contains 164 and 93, respectively.
(XLSX)

**S1 Fig. Absolute error distribution for order of COVID-19 discernible symptoms in 55,924 individuals in China [20].** The maximum error was determined from this distribution of absolute error and was used as a conservative measure of error to discern differences in transition probabilities of discernible symptom order.
(TIF)

**S2 Fig. Absolute error distribution for order of COVID-19 discernible symptoms in 373,883 individuals in the USA [17].** The maximum error was determined from this distribution of absolute error and was used as a conservative measure of error to discern differences in transition probabilities of discernible symptom order.
(TIF)

**S3 Fig. Relative change in number of searches of "Diarrhea" term per month in 2020 in USA.** A bar graph displaying the relative change, compared to the previous year, of search popularity of the term, "diarrhea", in the months, January, February, March, and April 2020 calculated from Google Trends [21].
(TIF)

**S4 Fig. The most likely path of discernible symptom order in regions characterized by the Wuhan reference strain (China [20] and Hong Kong [23]) and the D614G Variant (USA [17] and Brazil [24]).** A) The most likely paths of discernible symptom order using datasets from China and Hong Kong, which are both characterized by the Wuhan reference strain, are displayed. B) The most likely paths of discernible symptom order using datasets from the USA and Brazil, which are both characterized by the D614G variant, are displayed.
(TIF)

**S5 Fig. Absolute error distribution for order of COVID-19 discernible symptoms in 59 individuals in Hong Kong [23].** The maximum error was determined from this distribution of absolute error and was used as a conservative measure of error to discern differences in transition probabilities of discernible symptom order.
(TIF)

**S6 Fig. Absolute error distribution for order of COVID-19 discernible symptoms in at least 50,000 individuals in Brazil [24].** The maximum error was determined from this

distribution of absolute error and was used as a conservative measure of error to discern differences in transition probabilities of discernible symptom order.
(TIF)

**S7 Fig. Absolute error distribution for order of COVID-19 discernible symptoms in 244 individuals in Japan before the D614G mutation became prominent [27].** The maximum error was determined from this distribution of absolute error and was used as a conservative measure of error to discern differences in transition probabilities of discernible symptom order.
(TIF)

**S8 Fig. Absolute error distribution for order of COVID-19 discernible symptoms in 2,636 individuals in Japan after the D614G mutation became prominent [28].** The maximum error was determined from this distribution of absolute error and was used as a conservative measure of error to discern differences in transition probabilities of discernible symptom order.
(TIF)

**S9 Fig. The most likely orders of discernible COVID-19 symptoms in patients from Japan before [27] and after [28] the D614G mutation became prominent.** A) The first, second, and third most likely orders of discernible symptoms, with transition probabilities between symptoms of 244 COVID-19 patients in the Osaka Prefecture before the D614G mutation became prominent. B) The first, second, and third most likely orders of discernible symptoms, with transition probabilities between symptoms of 2,636 individuals with COVID-19 from Japan after the D614G mutation became prominent.
(TIF)

**S10 Fig. Absolute error distribution for order of COVID-19 discernible symptoms in 249 individuals in Shanghai, China [29].** The maximum error was determined from this distribution of absolute error and was used as a conservative measure of error to discern differences in transition probabilities of discernible symptom order.
(TIF)

**S11 Fig. Absolute error distribution for order of COVID-19 discernible symptoms in 463 individuals, 94% of which have at least one confirmed comorbidity, in Detroit, Michigan, in the USA [31].** The maximum error was determined from this distribution of absolute error and was used as a conservative measure of error to discern differences in transition probabilities of discernible symptom order.
(TIF)

**S12 Fig. Absolute error distribution for order of COVID-19 discernible symptoms in 393 individuals in New York, New York in the USA [30].** The maximum error was determined from this distribution of absolute error and was used as a conservative measure of error to discern differences in transition probabilities of discernible symptom order.
(TIF)

**S13 Fig. Absolute error distribution for order of COVID-19 discernible symptoms in 531 individuals from Atlanta, Georgia in the USA [32].** The maximum error was determined from this distribution of absolute error and was used as a conservative measure of error to discern differences in transition probabilities of discernible symptom order.
(TIF)

**S14 Fig. The likeliest paths of discernible symptom order in the USA overall and by age group** [17]**.** The most likely path of discernible symptom order for all patients in the USA dataset is shown on the left. All other columns are the most likely path of discernible symptom order by age groups from the overall set. The initial transition is written in green if it is consistent with the overall most likely path. The error of each implementation is written in purple.
(TIF)

**S15 Fig. The three likeliest paths of discernible symptom onset in individuals with COVID-19** [20] **and at least one comorbidity in China** [33]**.** The first (gold), second (silver), and third (bronze) likeliest paths representing the order of discernible symptom onset of 399 individuals with COVID-19 and at least one comorbidity in China. In the case where the gold and silver lines converge, they both follow the same path.
(TIF)

**S16 Fig. The three likeliest paths of discernible symptom onset in individuals with COVID-19 and at least one comorbidity the USA** [31]**.** The first (gold), second (silver), and third (bronze) likeliest paths representing the order of discernible symptom onset of 463 individuals with COVID-19 and at least one comorbidity in Detroit, Michigan. In the case where the gold and silver lines converge, they both follow the same path.
(TIF)

**S17 Fig. The three likeliest paths of discernible symptom onset in individuals with COVID-19 in China** [20] **and individuals with COVID-19 and at least one comorbidity in China** [33]**.** A) The first likeliest path (gold) of discernible symptom onset in China (green) and in China with at least one comorbidity (light green). B) The second likeliest path (silver) of discernible symptom onset in China (green) and in China with at least one comorbidity (light green). C) The third likeliest path (bronze) of discernible symptom onset in China (green) and in China with at least one comorbidity (light green). The error of the transition probabilities of the China dataset is 0.010, whereas the error of the transition probabilities of the China with comorbidities dataset is 0.013.
(TIF)

**S18 Fig. Absolute error distribution for order of COVID-19 discernible symptoms in 399 individuals with at least one comorbidity in China** [33]**.** The maximum error was determined from this distribution of absolute error and was used as a conservative measure of error to discern differences in transition probabilities of discernible symptom order.
(TIF)

**S19 Fig. The three likeliest paths of discernible symptom onset in individuals with COVID-19 in the USA** [17] **and individuals with COVID-19 and at least one comorbidity in the USA** [31]**.** A) The first likeliest path (gold) of discernible symptom onset in USA (purple) and in Detroit with at least one comorbidity (light purple). B) The second likeliest path (silver) of discernible symptom onset in USA (purple) and in Detroit with at least one comorbidity (light purple). C) The third likeliest path (bronze) of discernible symptom onset in USA (purple) and in Detroit with at least one comorbidity (light purple). The error of the transition probabilities of the USA dataset is 0.063, whereas the error of the transition probabilities of the USA with comorbidities dataset is 0.044.
(TIF)

**S20 Fig. The initial transition probabilities in individuals with COVID-19 and COPD, HIV, and cancer.** A) Table of raw frequency data specifying the number and the percentage of patients that experienced fever and cough in the datasets including patients with

comorbidities, COPD and HIV, in the USA and cancer in China and the USA. B) Abridged Hasse Diagram depicting the transition probabilities from no symptom to fever or cough in patients with COVID-19 and COPD in the USA [35]. The double arrow signifies that the transition probability of the most likely first symptom is more than double the transition probability second most likely first symptom. The error of the transition probabilities is 0.061. C) Abridged Hasse Diagram depicting the transition probabilities from no symptom to fever or cough in cases with COVID-19 and HIV in the USA [34]. The error of the transition probabilities is 0.031. D) Abridged Hasse Diagram depicting the transition probabilities from no symptom to fever or cough in cases with COVID-19 and cancer in China [36]. The error of the transition probabilities is 0.026. E) Abridged Hasse Diagram depicting the transition probabilities from no symptom to fever or cough in cases with COVID-19 and cancer in the USA [37]. The error of the transition probabilities is 0.030.
(TIF)

**S21 Fig. Absolute error distribution for order of COVID-19 discernible symptoms in 164 individuals with COPD in USA [35].** The maximum error was determined from this distribution of absolute error and was used as a conservative measure of error to discern differences in transition probabilities of discernible symptom order.
(TIF)

**S22 Fig. Absolute error distribution for order of COVID-19 discernible symptoms in 93 individuals with HIV in USA [34].** The maximum error was determined from this distribution of absolute error and was used as a conservative measure of error to discern differences in transition probabilities of discernible symptom order.
(TIF)

**S23 Fig. The top three likeliest paths of discernible symptom onset in individuals with COVID-19 and Cancer in China (green) [36] and the USA (purple) [37].** A) The first (gold), second (silver), and third (bronze) likeliest paths representing the order of discernible symptom onset of 205 COVID-19 patients with cancer in Hubei (green). In the case where the gold and silver lines converge, they both follow the same path. B) The first (gold), second (silver), and third (bronze) likeliest paths representing the order of discernible symptom onset of 423 COVID-19 patients with cancer in New York (purple). In the case where the gold and silver lines converge, they both follow the same path.
(TIF)

**S24 Fig. The most likely orders of discernible COVID-19 symptoms in patients from China [36] and the USA [37] with COVID-19 and cancer.** A) The first, second, and third most likely orders of discernible symptoms, with transition probabilities between symptoms of 205 COVID-19 patients with cancer in Hubei. The error of the transition probabilities is 0.026. B) The first, second, and third most likely orders of discernible symptoms, with transition probabilities between symptoms of 423 COVID-19 patients with cancer in New York. The error of the transition probabilities is 0.030.
(TIF)

**S25 Fig. Absolute error distribution for order of COVID-19 discernible symptoms in 205 individuals with cancer in China [36].** The maximum error was determined from this distribution of absolute error and was used as a conservative measure of error to discern differences in transition probabilities of discernible symptom order.
(TIF)

**S26 Fig. Absolute error distribution for order of COVID-19 discernible symptoms in 423 individuals with cancer in USA [37].** The maximum error was determined from this distribution of absolute error and was used as a conservative measure of error to discern differences in transition probabilities of discernible symptom order.
(TIF)

**S1 Appendix. Statistics and Demographics of Data.**
(PDF)

## Acknowledgments

We thank Dr. John C. Martin for helpful discussions and critical reading of the manuscript. We dedicate this work to his memory.

## Author Contributions

**Conceptualization:** Joseph R. Larsen, Margaret R. Martin, John D. Martin.

**Data curation:** Joseph R. Larsen, Margaret R. Martin, John D. Martin.

**Formal analysis:** Joseph R. Larsen, Margaret R. Martin.

**Funding acquisition:** Peter Kuhn.

**Investigation:** Joseph R. Larsen, John D. Martin.

**Methodology:** Joseph R. Larsen.

**Project administration:** James B. Hicks, Peter Kuhn.

**Resources:** James B. Hicks, Peter Kuhn.

**Software:** Joseph R. Larsen.

**Supervision:** James B. Hicks, Peter Kuhn.

**Validation:** Margaret R. Martin.

**Visualization:** Joseph R. Larsen, Margaret R. Martin, John D. Martin.

**Writing – original draft:** Joseph R. Larsen, Margaret R. Martin, John D. Martin.

**Writing – review & editing:** Joseph R. Larsen, Margaret R. Martin, John D. Martin, James B. Hicks, Peter Kuhn.

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
