## [Decision Letter · Decision Letter 0]

4 Jun 2021

Dear Dr. Kuhn,

Thank you very much for submitting your manuscript "Modeling the Onset of Symptoms of COVID-19: Effects of SARS-CoV-2 Variant and Patient Comorbidities" for consideration at PLOS Computational Biology.

As with all papers reviewed by the journal, your manuscript was reviewed by members of the editorial board and by several independent reviewers. In light of the reviews (below this email), we would like to invite the resubmission of a significantly-revised version that takes into account the reviewers' comments.

Apologies for the delay in coming to a decision. As you'll see, the reviewers were split on the importance of the study, and we were unable to secure a third reviewer despite many attempts. I think you may have some work to convince the negative reviewer, but I am willing to give you a chance at it. If this does not work out, then you may consider PLoS Digital Health instead.

We cannot make any decision about publication until we have seen the revised manuscript and your response to the reviewers' comments. Your revised manuscript is also likely to be sent to reviewers for further evaluation.

Sincerely,

Bard Ermentrout

Associate Editor

PLOS Computational Biology

Virginia Pitzer

Deputy Editor-in-Chief

PLOS Computational Biology

I think you may have some work to convince the negative reviewer, but I am willing to give you a chance at it. If this does not work out, then you may consider PLoS Digital Health instead.

Reviewer's Responses to Questions

**Comments to the Authors:**

Reviewer #1: This paper discusses the order of symptoms after SARS-CoV-2 infections. Using Hasse diagrams and large datasets of self-reported symptoms, comorbidities, and cancer status, the authors study how likely a set of four principal symptoms (fever, cough, nausea/vomiting, and diarrhea) are likely to occur and in which order. They expand on previous work in which they performed the same analysis on early data from Wuhan, before the emergence in the Summer of 2020 of the D614G mutation, which is now the globally-dominant variant. They report that fever and cough interchange as the first most likely symptom in those infected with SARS-CoV-2 without the D614G mutation and those infected with D614G. I found the methodology and analyses sound and the paper to be well-written. I do have a couple questions:

1) Line 251: The earlier dataset did not report nausea/vomiting, so they weren't included in the analyses of the Japanese data (which is of high interest given that the early data was taken before the emergence of D614G). What is the impact of this on predicted outcomes, since nausea/vomiting is in the third most likely path in the USA dataset, and is also found in the early China dataset? Limitations need be more clearly stated in the discussion.

2) Lines 269-279: The analysis of Google search terms is a weak point of the present paper, but does not affect the results. Given the prevalence of reporting on COVID-19, it seems impossible to untangle whether increases in specific search terms are related to increased symptoms or increases (global) reporting of symptoms in other countries. Again, limitations should be discussed.

3) Lines 445-446: Should be updated to reflect the variants of concern now in circulation.

Reviewer #2: Larsen et al. here analyzed the symptom data from thousands of SARS-COv2 infected patients in China, US and Japan. They modeled the order of symptom onset, and its potential association with virus mutations.

This is an interesting exercise, and could fit into a modeling journal, albeit the implications are somewhat limited. Clinically, the symptoms and their order are not specific enough to really help in diagnosis. Plus, in the US and Western world in general, there is a widespread access to testing, so I am not sure that this is really needed. This however could be more interesting in countries where access to testing is more limited, but in that case specific data to these countries should be analyzed. The association between symptom order and viral strains is potentially interesting but is very speculative in absence of individual data. Many factors could influence the symptom order (including seasonal change, population characteristics), not to mention the reliability and precision of such self-reported symptom to infer subtle changes in the order of symptom onset. This can nonetheless be an interesting modeling exercise, but much more attention should be given to the data and statistical aspects.

• Clarify where the data can be accessed to, how the data look like without any modeling. This is critical to ensure data transparency and reproducibility. I don’t see any basic statistics to give a sense of the data to the reader, nor a sense on how much the modeling exercise is reasonable in regard to the information available. How many individuals, their basic socio-demographic characteristics (age, BMI), what are the symptoms collected, for how long, the prevalence of each symptom over time since symptom onset, how many are pcr-confirmed, how many are antibody confirmed…

• Statistical aspects: very unclear how one path is preferred to the other, what are the statistical criterion associated with the decision inferred? As the paper stands, it is very difficult to understand the methodology, and how the statistical models have been chosen, as well as the confidence associated with each preferred path.

• The order of symptoms is important but probably even more important is the timing of these symptoms. Can you use the model to project on the timing of onset of these symptoms?

**Have all data underlying the figures and results presented in the manuscript been provided?**

Reviewer #1: Yes

Reviewer #2: **No: **please provide an access to the data set and/or a detailed description of the data used (see comments)

PLOS authors have the option to publish the peer review history of their article (what does this mean?). If published, this will include your full peer review and any attached files.

Reviewer #1: No

Reviewer #2: No
---

## [Decision Letter · Decision Letter 1]

17 Aug 2021

Dear Dr. Kuhn,

Thank you very much for submitting your manuscript "Modeling the Onset of Symptoms of COVID-19: Effects of SARS-CoV-2 Variant and Patient Comorbidities" for consideration at PLOS Computational Biology.

As with all papers reviewed by the journal, your manuscript was reviewed by members of the editorial board and by several independent reviewers. In light of the reviews (below this email), we would like to invite the resubmission of a significantly-revised version that takes into account the reviewers' comments.

We cannot make any decision about publication until we have seen the revised manuscript and your response to the reviewers' comments. Your revised manuscript is also likely to be sent to reviewers for further evaluation.

Sincerely,

Bard Ermentrout

Associate Editor

PLOS Computational Biology

Virginia Pitzer

Deputy Editor-in-Chief

PLOS Computational Biology

Reviewer's Responses to Questions

**Comments to the Authors:**

Reviewer #1: The authors have responded to my comments, thank you. In their revisions, the authors note

"To investigate these possible factors, we determined the most likely orders of symptom onset for various datasets from different cities in the USA (Detroit, Michigan, New York, New York, and Atlanta, Georgia)"

and performed similar analyses in both China and Japan. It is unclear to me whether data for these cities were included in the original country-level datasets? If so, these results are not surprising, and the analysis would need to be redone using independent data (validation step).

Reviewer #2: I thank the authors for their detailed response to my comments. Some of my initial comments have been well addressed, but I still have comments on the data description and statistical aspects for your consideration and that of the Editor.

• Thank you for adding the raw data. It appears from Table S1 that there is a huge heterogeneity in the data used. Data coming from cities, as well as data in individuals with comorbidities/cancer are very sparse. Since the results are not different than in the general population, I would recommend to put most of these analyses in the supplementary and to focus on the main message.

• Instead I would recommend to put the description of the data in the main manuscript (focusing on the data that are used in the main). Following my initial comment, it is very important that the model predictions do not appear as a black box. Even if PLoS CB is a modeling journal, readers need to know what information is immediately visible from the data, and which one is extracted from the model. I would therefore recommend to have a figure showing, for the main data used, the raw frequency data, focusing on the first and second symptoms. We need to get a sense of the data and not only the results.

• Regarding statistical aspects, I would recommend to systematically include confidence interval along with every point estimate of transition probability. This is particularly important given the heterogeneity in the population size used in the study.

• If I understand correctly lines 706-717, there is no formal statistical test to determine a type I error and how much confidence we have that one path is preferred to another. If there is no possibility to test the likelihood of symptom order (at least the two first one), this should perhaps be added in the discussion

• The public health implications should be clarified. I am not sure to understand the rationale for “support the practice of reporting the order of symptom onset from patients”. I don’t think that one would expect that this reporting could substitute or even complement viral sequencing. This is not a problem for a publication, the interest of the approach lies in its description of potentially different symptom order across viral strains, and the possibility that it may be one additional factor for increased transmission associated with D614G, but I don’t see the need to “oversell” the approach. Please consider modify these aspects in the abstract and in the discussion.

**Have the authors made all data and (if applicable) computational code underlying the findings in their manuscript fully available?**

Reviewer #1: **No: **The code was not provided

Reviewer #2: Yes

PLOS authors have the option to publish the peer review history of their article (what does this mean?). If published, this will include your full peer review and any attached files.

Reviewer #1: No

Reviewer #2: No
---

## [Decision Letter · Decision Letter 2]

10 Nov 2021

Dear Dr. Kuhn,

We are pleased to inform you that your manuscript 'Modeling the Onset of Symptoms of COVID-19: Effects of SARS-CoV-2 Variant' has been provisionally accepted for publication in PLOS Computational Biology.

Best regards,

Bard Ermentrout

Associate Editor

PLOS Computational Biology

Virginia Pitzer

Deputy Editor-in-Chief

PLOS Computational Biology

Reviewer's Responses to Questions

**Comments to the Authors:**

Reviewer #2: I thank the authors for acknowledging limitations on the precision of the estimates. As said originally the possibility that different strains may lead to different symptom order is interesting and could account for some differences in transmission rates. My comments on the data description remains to some extent, and I am still not sure to be able to evaluate to what extent reliable conclusions on symptom order can be obtained from data set that only contain prevalence data of each symptom, but no information on their sequential aspects. I recognize however that the method used in the paper is not in my area of expertise.

**Have the authors made all data and (if applicable) computational code underlying the findings in their manuscript fully available?**

Reviewer #2: **No: **please update the link to datasets. Some of them are no longer available (such as the first dataset referenced from WHO)

PLOS authors have the option to publish the peer review history of their article (what does this mean?). If published, this will include your full peer review and any attached files.

Reviewer #2: No

---

## [Editor Report · Acceptance letter]

23 Nov 2021

PCOMPBIOL-D-21-00279R2 

Modeling the Onset of Symptoms of COVID-19: Effects of SARS-CoV-2 Variant

Dear Dr Kuhn,

I am pleased to inform you that your manuscript has been formally accepted for publication in PLOS Computational Biology. Your manuscript is now with our production department and you will be notified of the publication date in due course.

With kind regards,

Olena Szabo
